# The Occurrence of Squalene in Human Milk and Infant Formula

**DOI:** 10.3390/ijerph191912928

**Published:** 2022-10-09

**Authors:** Aleksandra Purkiewicz, Sylwester Czaplicki, Renata Pietrzak-Fiećko

**Affiliations:** 1Department of Commodity Science and Food Analysis, Faculty of Food Sciences, University of Warmia and Mazury in Olsztyn, Cieszyński 1 Sq, 10-719 Olsztyn, Poland; 2Department of Food Plant Chemistry and Processing, Faculty of Food Sciences, University of Warmia and Mazury in Olsztyn, Cieszyński 1 Sq, 10-719 Olsztyn, Poland

**Keywords:** squalene content, bioactive ingredients, breastfeeding, lactation

## Abstract

(1) Background: The aim of the conducted research was to analyze the squalene content in infants’ food. (2) Methods: The experimental material included human milk collected from 100 women from Poland and three different infant formulas. The breast milk fat was extracted according to the Rose–Gottlieb method (AOAC), while the squalene content was determined using the high-performance liquid chromatography (HPLC) method. (3) Results: The highest amount of squalene was identified in the milk of women aged 18–25 (*p* < 0.05), and its content in milk decreased with the age of lactating women. Moreover, the greatest amount of squalene was identified in milk from the first lactation period (colostrum), while in mature milk, its content was more than two times lower. There was a correlation between breastfeeding BMI and the squalene content in milk (r = 0.78). (4) Conclusions: The conducted research shows that the level of squalene in human milk depends on physiological factors such as the lactation period and individual factors (age, BMI). The results of the conducted research indicate that breast milk is richer in squalene than modified milk. This study shows the importance of breastfeeding and indicates the superiority of breast milk over infant formulas.

## 1. Introduction

Mother’s milk is considered the gold standard in infant nutrition; due to its unique composition that adapts individually to the needs of the child, it is highly recommended that babies be fed only the mother’s milk until the child is at least 6 months old [1,2]. Such a position is recommended by, amongst others, the American Academy of Pediatricians (AAP) and the American Academy of Family Physicians (AAFP), who emphasize that breastfeeding reduces the risk of disease later in life [3]. Gregg et al. [4] indicate that lactation is the period in which the baby’s health is largely dependent on the mother’s milk, and its ingredients influence the baby’s metabolic programming. In addition to the appropriate amount of energy and nutrients needed for the proper growth of a young organism, it also contains several functional ingredients with a bioactive effect [5]. These ingredients significantly affect the health of the newborn, stimulate the immune system, and protect against infections in the body [6]. Among the bioactive components of human milk are proteins, oligosaccharides, polyunsaturated fatty acids (PUFA), fat-soluble vitamins, hormones, and growth factors or probiotics [7]. Lipids play a special role in the development of a young organism. During the first month of life, a newborn child derives energy from fats at the level of about 50%. Therefore, during this period, in addition to the source of energy, lipids are a very important building function of the brain and nervous system [8]. An important component of human milk is squalene [9]. Squalene is a triterpene hydrocarbon (C_30_H_50_) with six double bonds in its structure [9]. It is a non-steroidal cholesterol precursor; it is absorbed from the intestine and synthesized in adipose tissue, liver, and skin [10]. This relationship in the human body is responsible for several processes. Squalene is synthesized into acetyl-CoA, is a precursor of biologically important compounds, stimulates the immune system, and protects the skin against UV radiation [9]. Squalene is reported to have several bioactive properties, including antioxidant, anti-cancer, and anti-atherosclerotic properties [11,12].

Piesiewicz [13] reports that squalene constitutes 0.12% of the lipid composition of human milk, which is about 2–6% of the total cholesterol concentration in breast milk [14]. Squalene is necessary for the development of the body’s immunity [15]. This is important for both the mother and the baby in terms of the health-promoting properties of squalene. This component has a beneficial effect on the immune system and the acquisition of natural human immunity, which enables the support of the underdeveloped immune system of the child. In addition, it acts as an antidote in the body by removing poisons, mainly xenobiotics, which are sometimes found in breast milk fat. Many poisons can dissolve into squalene and thus are removed from the body along with feces [13].

The human body synthesizes some amounts of squalene on its own. This ingredient plays a significant role in the human body—100 g of sebum contains about 30 mg of squalene. It is also found in the skin (14.8 mg/100 g), thyroid gland (5.5 mg/100 g), lymph glands (5.2 mg/100 g), the inner layer of arteries (4.0 mg/100 g), and the spleen (3.0 mg/100g) [13]. High levels of squalene are also found where the infection-fighting white blood cells are formed. In addition, squalene is a precursor to cholesterol, steroid hormones, vitamin D3, and bile acids, which are essential compounds for human health [16]. The richest source of squalene is shark liver, which contains up to 70% of this compound. Sources of squalene among food products include vegetable oils such as amaranth (6300 mg/100g) and olive oil (685.5 mg/100 g), corn (27.9 mg/100 g), parmesan cheese (9.6 mg/100 g), and butter (6.1 mg/100 g) [13].

So far, it has been shown that squalene is an important component that determines the proper development of a child. Despite the proven presence of squalene in human milk, only a few authors have undertaken a study of the content of this component in breast milk. Moreover, there are no studies to indicate the content of squalene in modified milk in the presently available literature. The literature shows a relationship between the decline in squalene synthesis with age and its variable content in milk depending on the lactation period [14,17]. A novelty in the undertaken research is the analysis of the squalene content in relation to age, lactation period, and BMI in human milk. This research aimed to analyze the squalene content in human milk and infant formulas.

## 2. Materials and Methods

### 2.1. Chemicals

Ethyl alcohol, ammonia, diethyl ether, hexane, acetonitrile, isopropyl alcohol, anhydrous sodium sulfate, and squalene standard were obtained from Sigma Aldrich (St. Louis, MO, USA).

### 2.2. Research Material

The research material consisted of samples of human milk and modified milk from three producers. Samples of human milk were collected from 100 women who are natives and presently live in the east–northeast region of Poland. The inclusion criteria for the study included the proper health of women, no contraindications to breastfeeding, and easy delivery. The surveyed women were asked to complete a questionnaire containing questions about personal data (first name, surname, age, place of residence), body weight and height, current diseases, medications used frequently, and information on pregnancy and lactation (type of childbirth, date of birth of the child, date of milk sampling, child’s sex, weight and height of the child, and the number of children born). Based on a nutritional interview with women, none of them included the characteristic food products that are sources of squalene (amaranth, olive oil, fish oils). The study group included women aged 18–44 years. The breastfeeding population was organized into age categories (18–25 years old, 26–33 years old, and 34–44 years old), by stages of lactation (1st—colostrum: breast milk until a few days after birth, 2nd—foremilk: breast milk until two weeks after birth, 3rd and 4th: hindmilk—breast milk from 4 weeks of breastfeeding), and BMI (<18.5 kg/m^2^—underweight; 18.5–24.99 kg/m^2^—normal weight; 25.0–29.99 kg/m^2^—overweight; >30 kg/m^2^—obese) [18]). The characteristics of the studied population group are presented in Table 1. The milk samples (50 to 100 mL) were collected from each woman using electronic breast pumps immediately after breastfeeding into sterile glass bottles. If it was not possible to collect the right amount of milk during one breastfeeding session (50 to 100 mL), the women were asked to store the milk in a tightly closed container in the refrigerator (4 ± 1 °C), and to refill it during the next breastfeeding session. The milk samples were stored in special containers maintaining the temperature, and then frozen and stored at a temperature of −30 °C. The research material also included modified milk from three producers intended for the initial feeding of infants. Modified milk was purchased on the Olsztyn market. Each of the milk samples was milk modified on the basis of cow’s milk. The tested modified milks were characterized by a similar energy value and nutritional value. The modified milk was prepared for analytical tests in accordance with the instructions on the manufacturer’s packaging.

### 2.3. Sample Extraction

The human milk samples were removed from the freezer, thawed, and mixed rapidly. The tested breast milk samples were heated to 40 ± 1 °C. A process of fat extraction from the breast milk was performed according to the Rose–Gottlieb method [19]. Approximately 10 g of milk (weighed with an accuracy of 0.01 g) was added to 2 mL of 10% ammonia and 10 mL of ethyl alcohol, each time gently mixing. From the mixture prepared in this process, the fat was extracted with diethyl ether and hexane in an amount of 25 mL each. The upper organic layer was collected and then dried by filtration through anhydrous sodium sulfate. The organic solvents were evaporated using a rotary evaporator. A double extraction of milk fat was used.

### 2.4. Determination of Squalene Content Using High-Performance Liquid Chromatography Method (HPLC)

The determination of the squalene content in the sample was carried out using the high-performance liquid chromatography (HPLC) technique according to the method developed as a result of the modification of the procedure for the determination of triacylglycerols described by Czaplicki et al. [20].

A sample of the women’s milk fat (about 200 mg) was transferred into a 10 mL volumetric flask, filled up with n-hexane, and purified by spinning. The analysis was carried out using a 1200 series liquid chromatograph manufactured by Agilent Technologies (Palo Alto, CA, USA), equipped with a DAD detector (set at 218 nm) from the same manufacturer. Chromatographic separations were conducted on a Merck LiChrospher RP-18 column, 250 mm × 4.6 mm, 5 mm, at 30 °C. A gradient system was used for elution: A, acetonitrile; B, isopropyl alcohol; C, hexane. The elution profile was 0–12 min, 20–22% B and 10–12% C in A (linear gradient), 12–15 min 22–25% B and 12–25% C in A (linear gradient), 15–20 min, 25% B and 25% C in A (isocratic), 20–25 min, 25–20% B and 25–10% C in A (linear gradient). The mobile phase flow rate was 1 mL/min.

The external calibration curve was used for quantitative analysis. The squalene standard was supplied by Sigma-Aldrich. The calibration curve is described by an equation: squalene peak area = 27.23023 × amount (mg/L) + 1.57784 with linearity proved by a correlation coefficient 0.9997. The detection limit for the method was determined as 62.5 µg/100 g of milk lipids and the quantitation limit was 208.3 µg/100 g of milk lipids.

### 2.5. Statistical Analysis

Descriptive statistics were used to characterize the studied population group. The obtained results were presented as mean ± standard deviation (SD). The normal distribution was determined with the Shapiro–Wilk test, while the homogeneity of variance was determined with Levene’s test. The studied distribution was not normal and the variance was not homogeneous, thus, non-parametric tests were used for the analyses. The Kruskal–Wallis and Dunn’s post hoc test compared the quantitative data between the distinguished groups by age and stage of lactation. Linear Spearman’s correlation coefficients were calculated to show the relationships between BMI (body mass index) and the content of squalene in the breast milk samples. The significance level was α ≤ 0.05. The program Statistica 13.1 (Statsoft Inc., Tulsa, OH, USA) was used for the statistical analysis.

## 3. Results

The squalene content was determined in the tested samples of human milk and modified milk from selected producers. The results of the squalene content in the tested samples are summarized in the graphs (Figure 1, Figure 2 and Figure 3) and Table 2. 

Squalene was quantified in 67% of the human milk samples (*n* = 67), while 33% contained <0.208 mg squalene per 100 g fat, which is the limit of quantification of the tested component. Figure 1, Figure 2 and Figure 3 and Table 2 take into account the squalene content in the milk samples of 67% of the women, in which its content could be analytically determined. Figure 1 shows the squalene content in the human milk samples as a function of the age of the women. There were significant differences in the squalene content by age. The highest content of squalene was found in the milk of women aged 18–25 (*p* ≤ 0.05) and lowered with the age of women. The milk of women aged 26–33 years contained 1.7 times less squalene than the milk of women aged 18–25, while the milk of women aged 34–42 years contained almost 2.5 times less (*p* ≤ 0.05). Significant differences in the content of squalene in human milk according to the lactation period were also noted. According to the research, the milk from the first lactation period contained the most squalene, while the milk from the third lactation period had the least (*p* ≤ 0.05). The milk from the first lactation period contained 1.5 and more than 2 times the squalene than the milk from the second and third lactation periods, respectively.

In Figure 3, the relationship between the content of squalene in breast milk and the BMI of the studied women was analyzed. It was shown that as the BMI increased, the women’s milk contained more squalene. The lowest squalene content was recorded in the milk samples of underweight women (BMI < 18.5 kg/m^2^), while the highest was found among obese women (BMI > 30 kg/m^2^). There was a significant correlation between the BMI of breastfeeding women and the content of squalene in breast milk (r = 0.78).

Table 2 shows the average content of fat, cholesterol, and squalene in human milk and the modified milk of selected producers. The human milk contained an average of 4 g of fat per 100 mL, while the fat content of formula milk ranged from 3.1 to 3.7 g/100 mL. In the case of cholesterol, based on the literature data, it has been shown that its content in human milk is, on average, 16.5 mg/100 mL, while in modified milk A, B, and C it is successively almost 7, 3, and almost 18 times lower, respectively. The cholesterol content in human milk and infant formulas correlated with the content of squalene. A higher squalene content in breast milk results in a higher cholesterol content. The average level of squalene in breast milk was 17 mg/100 g fat, while the modified milk contained only traces of squalene (<0.208 mg/100 g fat); in the case of cholesterol, the breast milk contained values several times higher than the modified milk.

## 4. Discussion

This research presents the level of squalene in baby food—human milk and formula—and shows the influence of selected factors (age of breastfeeding women, lactation period, BMI) on the level of squalene in breast milk.

The course of a normal pregnancy is associated with an increased level of cholesterol in the blood serum in women, therefore the level of cholesterol precursors, namely, squalene, desmosterol, and lanosterol, during lactation are also at a higher level [22]. Pregnancy and lactation are closely related to increased cholesterol synthesis, which is essential for the proper development of the newborn [23]. Of the breast milk samples tested, 67% of them contained squalene, while 33% of the milk samples contained <0.208 mg squalene per 100 g fat, which is considered to be below the limit of detection. The squalene content varies according to many factors. The group of 33% of women whose milk samples did not contain squalene mainly consisted of women over 30 with a BMI within the normal range. However, in almost 20 women, the BMI value was at the lower limit of the norm (18.5–19.2 kg/m^2^). Consequently, for the percentage of lactating women whose milk samples did not contain a measurable amount of squalene, it was not possible to determine the exact squalene content in these samples. The amount of squalene in the body also changes with age. This ingredient is produced in the greatest amounts until the age of 25. After this time, its concentration gradually decreases and the body is more exposed to oxidative stress [17]. With age, the production of squalene in the body decreases, and at the same time, the demand for it begins to increase. Newborn babies have the highest level of squalene, and after the age of 30, their production begins to drop rapidly [24]. Since a positive correlation has been demonstrated between the presence of squalene in the blood serum of the mother and the infant and its presence in the mother’s milk, it’s assumed that with the age of lactating women, consequently the content of squalene in the mother’s milk also decreases. Even though squalene is present in breast milk in small amounts, it has a strong effect on free oxygen radicals, thus stimulating the immune system [25]. Pertaining to infants and young children, an important function of squalene is its detoxifying function, including the ability to bind and block pesticides, xenobiotics, dioxins, and other poisons that pass from the contaminated environment into the body and then into breast milk.

The composition of breast milk varies during the lactation period, mainly in terms of dry matter, non-fat dry matter, fat, and protein [26]. Differences in the composition of breast milk concerning the lactation period are particularly noticeable in the fat fraction of milk. In addition to the total content of fat, its composition also changes. One of the components of the fat fraction of milk—the amount of which depends on the lactation period—is squalene. Kallio et al. [14] showed that the concentration of squalene increased between the second and sixth month of lactation (from 9.4 to 12.0 µmol/L), while it decreased between the sixth and ninth month of lactation (from 12.0 to 11.0 µmol/L). In turn, Laitinen et al. [27] reported that the concentration of squalene in breast milk decreased between the first and third trimester of pregnancy (by more than three units), and increased again after delivery. Moreover, serum squalene levels were also assessed in the children of the mothers studied and it was shown that the level of squalene decreased between 1 and 6 months of age (from 36.5 to 27.2 mmol/mmol cholesterol). Mothers in different periods of lactation participated in the study. The greatest amount of squalene was contained in colostrum (first lactation period), and its content was more than two times lower in mature milk (third lactation period).

In the conducted research, an analysis of the relationship between squalene level in breast milk and BMI was undertaken. Miettinen et al. (2014) indicated increased cholesterol synthesis among obese women with BMI > 30 kg/m^2^. In the studies undertaken, along with the increase in the BMI of the lactating women, an increased presence of squalene in the breast milk was also observed. Squalene is one of the precursors of cholesterol, and obesity in pregnancy and lactation is often associated with elevated markers of serum cholesterol synthesis, and because the fat components including squalene can penetrate breast milk [15], its increased presence in the blood may also result in its increased content in breast milk. Overweight and obesity very often correlate with an increased level of adipose tissue. Adipose tissue synthesizes and stores large amounts of squalene, and squalene is transported in triglyceride-rich lipoproteins. Consequently, a higher BMI may lead to higher levels of squalene in breast milk. Moreover, it should be emphasized that overweight and obesity, and rapid changes in body weight, have an impact on cholesterol metabolism [10]. Therefore, in people with excess body weight, the squalene distribution is higher and the body may generate a need for a higher squalene requirement [28]. In the group of studied women, the level of squalene in milk increased with an increase in BMI, which may indicate its higher synthesis in people with excess body weight.

Another subject of the research was the modified milk of various producers, intended for the initial feeding of infants. Formula milk mixtures were created as an alternative to mother’s milk. More attention is paid to the fact that the composition of the modified milk resembles mother’s milk as much as possible, but despite the improved production technology, the fortification of milk with some bioactive ingredients is impossible [29]. The critical difference between the composition of human milk and modified milk is the fat fraction. A unique ingredient in breast milk is long-chain polyunsaturated fatty acids, including docosahexaenoic acid (DHA), necessary for the proper development of the brain, nervous system, and eyesight of the child [30]. While the fortification of modified milk with DHA has been mandatory in the European Union since 2020 [31], there is still a lot of emphasis on the advantages of possible long breastfeeding. This is because modified milk is enriched with artificial omega-3 fatty acids, and does not contain alkylglycerols and squalene, which can support the infant’s immune system, fighting not only against infections, but also protecting against various chronic diseases [15]. In the tested modified milk from different producers, the squalene content was lower than the lower detection limit of this component in milk fat. Based on the tested samples of infant formula, it was found that they were not a rich source of squalene. However, for a more detailed analysis, it would be necessary to test more samples of modified milk from different producers in order to reach a clear conclusion on the squalene content in infant formula. Fat is an ingredient from which a newborn obtains energy at the level of about 50%, so it is extremely important to provide infants with the best-composed food. Breast milk adapts individually to the needs of the child. Moreover, unlike modified milk, it contains immune bodies that shape the immature immune system. Polyunsaturated fatty acids and squalene are components that pass from the mother’s body into breast milk. Therefore, breastfeeding ensures that an adequate level of such compounds is available for the child during infancy and early age. Moreover, it prevents deficiencies and the resulting autoimmune disorders (rheumatoid arthritis, psoriasis, Hashimoto’s disease, and type 1 diabetes) [15]. An important component of breast milk is cholesterol. It is present in very little or no form in the formula, hence the absence of squalene and other cholesterol precursors. For a newborn child, contact with cholesterol from the mother’s milk is extremely important because it teaches the child’s body to properly manage it and avoid metabolic problems later in life [32].

Omega-3 fatty acids, squalene, and alkylglycerols can pass into breast milk. At the stage of pregnancy and breastfeeding, an appropriate diet is particularly important as it will determine the correct level of these compounds for the child’s body [15]. The levels of squalene in the breast milk of lactating women, differing in terms of age, lactation period, and BMI, did not result from a varied diet. Even though squalene can be supplemented with the diet, the women did not receive dietary recommendations regarding the inclusion of squalene-rich products in their diet, such as olive oil, shark liver oil, or amaranth [33], and did not take dietary supplements containing squalene.

Given that the many bioactive ingredients in modified milk do not include squalene, it is recommended that infants be fed exclusively with mother’s milk up to the age of 6 months, and continue to be given it at least until the age of 24 months. This method of feeding infants ensures that the child is provided with all of the necessary nutrients and ensures the best development of the young organism. It should be emphasized that despite squalene having proven health-promoting properties, there is still little research on the content of this nutrient in breast milk and its health-promoting properties on a child. The conducted studies have shown that the content of squalene in breast milk varies depending on the mother’s age, lactation period, and body weight and its relation to body height (BMI). Although the majority of breast milk samples contained squalene, 33% of them still contained squalene to an unmeasurable value (below the lower limit of detection of squalene in fat). Although the World Health Organization recommends breastfeeding as long as possible and emphasizes the high nutritional value of mother’s milk due to the unique composition (the content of proteins, polyunsaturated fatty acids, oligosaccharides, immune factors, enzymes, and hormones), so far, no unambiguous standards have been established regarding the content of cholesterol precursors, including squalene, in breast milk and the resulting pro-health properties. 

## 5. Conclusions

Breastfeeding allows you to provide the young body with a number of nutrients necessary for the proper functioning of the body. More squalene was identified in women aged 18–25 compared to older mothers, and in colostrum compared to foremilk and hindmilk. Finally, a relationship between the content of squalene in milk and BMI has been demonstrated: the squalene content increased with BMI growth. The tested samples of modified milk did not contain any measurable content of squalene. In the study, a third of the human milk samples, like the samples of infant formula, contained squalene at the level of <0.208 mg/kg fat. Nevertheless, on the basis of the research carried out, breast milk can be considered a better source of squalene compared to modified milk.

## Figures and Tables

**Figure 1 ijerph-19-12928-f001:**
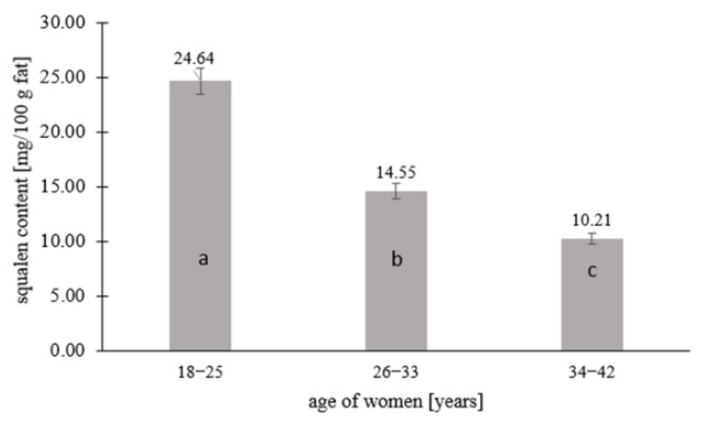
Squalene content in human milk in relation to the age of lactating women. Means followed by a different letter (a, b, c) are significantly different (*p* < 0.05).

**Figure 2 ijerph-19-12928-f002:**
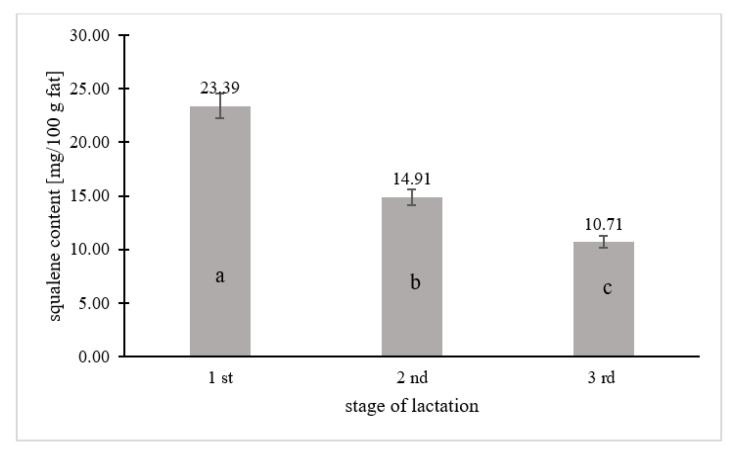
Squalene content in breast milk in relation to the lactation period. Means followed by a different letter (a, b, c) are significantly different (*p* < 0.05).

**Figure 3 ijerph-19-12928-f003:**
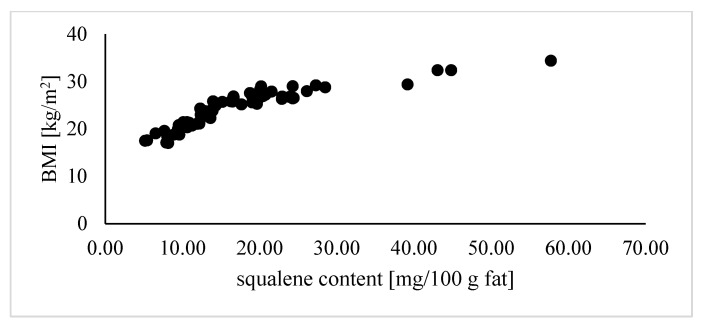
Correlation coefficients (r) calculated for the relationship between the BMI of lactating women and squalene content in human milk (correlation: r = 0.78).

**Table 1 ijerph-19-12928-t001:** Characteristics of the studied breastfeeding women.

Factors	*n* = 100
Age, year	
18–25	32
26–33	39
34–44	29
Stage of lactation	
1st	33
2nd	39
3rd and 4th	28
BMI, kg/m^2^	
<18.5 (underweight)	9
18.5–24.99 (normal weight)	52
25.0–29.99 (overweight)	34
>30 (obese)	5

**Table 2 ijerph-19-12928-t002:** Average content of fat, cholesterol, and squalene in human milk and formula milk of selected producers.

Sample		Fat	Cholesterol	Squalene
		(g/100 mL) Mean (Range)	(mg/100 mL) Mean (Range)	(mg/100 g Fat) Mean (Range)
Human milk		4.0 (3.0–5.0)	16.50 (10.0–23.0)	16.93 (5.43–57.73)
Infant formula	A	3.4 (3.2–3.6)	2.48 (1.1–6.7)	<0.208
	B	3.3 (3.1–3.5)	5.45 (4.6–6.5)	<0.208
	C	3.5 (3.2–3.7)	0.93 (0.4–1.3)	<0.208

Source: own study based on Kamelska et al. [21] and own research.

## Data Availability

The data presented in this study are available on reasonable request from the corresponding author.

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
