# Peer review of "The Occurrence of Squalene in Human Milk and Infant Formula"

_ijerph, 2022, doi:10.3390/ijerph191912928_

Round 1

Reviewer 1 Report

THE RESEARCH TACKLES AN INTERESTING SUBJECT, HOWEVER, IN MY OPINION ITS SERIOUS FLAW IS THE EXPERIMENTAL DESIGN, I.E. THE FACT THAT THE NUTRITIONAL HABITS OF THE LACTATING MOTHERS WERE NOT TAKEN INTO CONSIDERATION. AS THE AMOUNT OF SQUALENE IN MILK CAN VARY ACCORDING TO THE DIET, HOW CAN THE AUTHORS EXCLUDE THE IMPACT OF MOTHERS' DIET ON THE RESULTS OBTAINED?

L31-32. PLEASE, CLARIFY PHRASE.

L46 VS L60- IF SQUALENE IS ABSORBED, HOW CAN IT BE REMOVED BY FECES? PLEASE, RECONSIDER

L62: PLEASE, REPLACE "SKIN FAT" BY "SEBUM"

L110: PLEASE, INDICATE TEMPERATURE

L298-L300- THESE SHOULD NOT BE PART OF THE CONCLUSION

L103 VS L301-302: "The milk samples (50 to 100 mL) were collected immediately after feeding the baby 103 with the use of an electronic breast pump."- THERE IS AN APPARENT CONTRADICTION REGARDING THE TIMING OF MILK COLLECTION- PLEASE, RECONSIDER AND CORRECT THROUGHOUT ALL THE RELEVANT PARTS OF THE PAPER.

Author Response

Review 1

We would like to thank the reviewer for careful reading of the manuscript titled: 
The occurrence of squalene in human milk and infant formula.
We would like to thank the reviewer for their contributions and valuable comments which make it possible to improve our manuscript.

  1. THE RESEARCH TACKLES AN INTERESTING SUBJECT, HOWEVER, IN MY OPINION ITS SERIOUS FLAW IS THE EXPERIMENTAL DESIGN, I.E. THE FACT THAT THE NUTRITIONAL HABITS OF THE LACTATING MOTHERS WERE NOT TAKEN INTO CONSIDERATION. AS THE AMOUNT OF SQUALENE IN MILK CAN VARY ACCORDING TO THE DIET, HOW CAN THE AUTHORS EXCLUDE THE IMPACT OF MOTHERS' DIET ON THE RESULTS OBTAINED?

Response:

Thank you for your valuable comment. In the undertaken study, the influence of the diet on the content of squalene in human milk was not analyzed for several reasons. First, it has been analyzed from the literature that squalene occurs naturally in human milk [https://www.researchgate.net/publication/20353827_Cholesterol_and_its_precursors_in_human_milk_during_prolonged_exclusive_breast-feeding; https://digital.csic.es/bitstream/10261/245588/1/Composition_Goudjil_PV_Art.pdf]; https://www.jlr.org/article/S0022-2275(20)37471-X/pdf), and the aim was to investigate its variability in milk depending on factors such as mother's age and weight body and height and the resulting BMI and the lactation period in which the women are. Based on a nutritional interview with women, none of them included the characteristic food products that are sources of squalene (amaranth, olive oil, fish oils). Therefore, no attempt has been made to include these products in women's diets as they are tastes that may not necessarily be acceptable to nursing mothers. In addition, there are still discussions about the effect of intense flavored foods on the taste of breast milk. Some foods have been shown to be able to significantly alter the effects of human milk (https://www.journalofdairyscience.org/article/S0022-0302(18)31132-9/fulltext), but for fish oil this is not the same (https: //www.academia.edu/25764271/Sensory_and_molecular_characterisation_of_human_milk_odour_profiles_after_maternal_fish_oil_supplementation_during_pregnancy_and_breastfeeding; https://hal.archives-ouvert24go. women (https://www.ncbi.nlm.nih.gov/books/NBK501898/). Although some sources state that the consumption of fish oil by lactating women does not affect the taste of breast milk (https://www.ncbi.nlm.nih.gov/books/NBK501898/), foods with intense flavors are reported to , may affect the taste of women's milk and adversely affect its sensory properties, which may make the food unacceptable / less acceptable to a child (https://www.scielo.br/j/ean/a/cpq8wSd79qvQNZkzj4D8XQK/?lang=en&format) = pdf). In the conducted research, it was not wanted to affect the comfort of the mother and the baby and disrupt standard feeding. When it comes to supplementation, its inclusion in the diet of pregnant and lactating women is debatable. Ingredients that are recommended for supplementation during lactation are folic acid, vitamin D, iodine, omega acids (DHA) and zinc; supplementation of other ingredients is not recommended in the population of healthy women (Recommendations of the Polish Gynecological Society on the use of vitamins and micronutrients in women planning pregnancy, pregnant and lactating women. Ginekol Pol. 2014; 85: 395–399), therefore no attempts were made to supplement breastfeeding women preparations containing squalene. Pregnant and lactating women are a particularly vulnerable and specific group and experimental studies carried out on this group must be carried out under strict supervision. Therefore, in the undertaken studies, it was not decided to include products rich in squalene in the diet of nursing mothers and only its variable content was examined in relation to the above-mentioned factors (age, BMI, lactation period). Nevertheless, thank you for your right attention and in the next research on the content of squalene, we will carefully consider the issue of nutrition for lactating women and try to examine the impact of diet on the formation of squalene content in human milk, without adversely affecting the comfort of mother and child.)

  1. L31-32. PLEASE, CLARIFY PHRASE.

Response:

L 31-32 is corrected. American Academy of Family Physicians (AAFP) instead of American Academy of Physicians (AAFP).

  1. L46 VS L60- IF SQUALENE IS ABSORBED, HOW CAN IT BE REMOVED BY FECES? PLEASE, RECONSIDER

Response:

Squalene, as a result of being a lipid, has strong detoxifying properties and helps remove poisons from the organism that get from the contaminated environment - xenobiotics or pesticides, among others. It is more difficult for the body to get rid of fat-soluble poisons than water-soluble ones. Squalene has a detoxifying effect on fat-soluble poisons, compounds such as xenobiotics or dioxins are easily dissolved in it, so they are removed from the body.

  1. L62: PLEASE, REPLACE "SKIN FAT" BY "SEBUM"

Response:

In line 62 „skin fat”is replaced by „sebum”

  1. L110: PLEASE, INDICATE TEMPERATURE

Response:

In line 110 informations about temperature were added. The tested breast milk samples were heated to 40 ± 1 º C.

  1. L298-L300- THESE SHOULD NOT BE PART OF THE CONCLUSION

Response:

 Lines 298-300 were deleted from conclusions.

  1. L103 VS L301-302: "The milk samples (50 to 100 mL) were collected immediately after feeding the baby 103 with the use of an electronic breast pump."- THERE IS AN APPARENT CONTRADICTION REGARDING THE TIMING OF MILK COLLECTION- PLEASE, RECONSIDER AND CORRECT THROUGHOUT ALL THE RELEVANT PARTS OF THE PAPER.

Response:

Thanks for your comment. I agree that milk sampling was not entirely clearly described by the authors and some contradictions emerged. In the Materials and Methods section, the section concerning milk samples has been corrected.

Reviewer 2 Report

The submitted manuscript presented a study to investigate the squalene content in breast milk and selected infant formulas. Squalene is potentially a nutrient that maybe beneficial to the infant development. Various breast milk were analyzed to provides insight into the difference of squalene content in different population groups. However, I have several major concerns about this manuscript.

1.      Section 2.4, the analysis of squalene was described in brief. I’m uncertain the simple HPLC separation employed in this study would be able to separate squalene in the presence of other milk fats in the complex samples. Was the analytical method validated to show acceptable accuracy and precision in analyzing milk fat samples? Please provide a chromatogram showing the squalene standard and the squalene in milk fat sample.

2.      Section 2.4, “The detection limit for the method was determined as 62.5 μg /100 g of milk lipids and the quantitation limit was 208.3 μg / 100 g of milk lipids.” Please elaborate how the LOD and LOQ were determined in this study.

3.      Three infant formulas were analyzed for squalene content in this study. Please provide more details about the infant formulas chosen to use in this study. Are they bovine milk-based formula or soy-based formula?

4.      Section 3, “…while 33% contained <0.208 mg squalene per 100 g fat”. It implied ~1/3 of breast milk samples did not contain measurable amount of squalene. Please explain which population groups these breast milk sample belong to. And why this data was not included in statistical analysis presented in Figure 1 to Figure 3?

5.      Page 7, line 270. “…it can be concluded that formula milk is not a rich source of squalene, regardless of the manufacturer.” It is arbitrary to make this conclusion by only analyzing three infant formula samples. A more in-depth analysis of squalene in infant formula is needed in order to make this claim. 

6.  Page 8, line 304. “The conducted studies indicate the superiority of human food over modified milk due to only trace amounts of squalene in infant formulas.” I disagreed with this statement. Firstly, I’m not aware of any scientific consensus about squalene as the essential nutrient in baby food and their recommended daily uptake. Moreover, as indicated in this study, the level of squalene in breast milk is highly variable, with 1/3 of samples contained squalene below LOQ. So it is improper to claim breast milk as better baby food because it contain highly variable amount of squalene that has unknown nutritional benefits.

Author Response

Review 2

We would like to thank the reviewer for careful reading of the manuscript titled: 
The occurrence of squalene in human milk and infant formula.
We would like to thank the reviewer for their contributions and valuable comments which make it possible to improve our manuscript.

  1. Section 2.4, the analysis of squalene was described in brief. I’m uncertain the simple HPLC separation employed in this study would be able to separate squalene in the presence of other milk fats in the complex samples. Was the analytical method validated to show acceptable accuracy and precision in analyzing milk fat samples? Please provide a chromatogram showing the squalene standard and the squalene in milk fat sample.

Response:

Quantification was achieved by injection of squalene solutions of known concentrations ranging from 0.05 to 40 µg/mL (R2 ≤ 0.9997). The method used was suitable for the analysis of squalene in milk samples as it is shown on chromatograms provided.

Fig. 1 Squalene standard

Fig. 2 Milk sample

Fig. 3 Milk sample and squalene standard

  1.  The detection limit for the method was determined as 62.5 μg /100 g of milk lipids and the quantitation limit was 208.3 μg / 100 g of milk lipids.” Please elaborate how the LOD and LOQ were determined in this study.

Response:

Limit of Detection (LOD) and Limit of Quantitation (LOQ) were determined using based on signal-to-noise ratio method. The mean value of the signal-to-noise ratio (n = 4) generated from the solution that just caused more than 3 times S/N ratio was used to calculate the detection limit (based on S/N = 3) and quantitation limit (based on S/N = 10) of squalene.

  1. Three infant formulas were analyzed for squalene content in this study. Please provide more details about the infant formulas chosen to use in this study. Are they bovine milk-based formula or soy-based formula?

Response:

 In the conducted study, 3 modified milks intended for initial feeding of infants were tested. Each of the milk samples was milk modified on the basis of cow's milk. The tested modified milks were characterized by a similar energy value and nutritional value.)

  1.  “…while 33% contained <0.208 mg squalene per 100 g fat”. It implied ~1/3 of breast milk samples did not contain measurable amount of squalene. Please explain which population groups these breast milk sample belong to. And why this data was not included in statistical analysis presented in Figure 1 to Figure 3?

Response:

Thank you for your valuable comment, you are right that the authors provided too little information on this issue in the Material and Methods section. The population of women (33%), in which milk was not identified with squalene, were mainly women over 30 with BMI within the normal range, but in almost 20 women the BMI value was at the lower limit of the norm (18.5-19.2). Consequently, the proportion of breastfeeding women whose milk samples did not contain measurable squalene content, the authors did not obtain specific values for the squalene content in these samples, and therefore the results could not be included in the statistical analysis. The authors in the "Results" section will further indicate that Figs 1-3 relate to 66% of lactators whose milk contained measurable squalene.

  1. Page 7, line 270. “…it can be concluded that formula milk is not a rich source of squalene, regardless of the manufacturer.” It is arbitrary to make this conclusion by only analyzing three infant formula samples. A more in-depth analysis of squalene in infant formula is needed in order to make this claim.

Response:

We agree with the Reviewer's comment. It has been too arbitrary written that infant formula is not a source of squalene if only samples from three producers were tested. The request has been modified. “Based on the tested samples of infant formula, it was found that they were not a rich source of squalene. However, for a more detailed analysis, it would be necessary to test more samples of modified milk from different producers in order to reach a clear conclusion on the squalene content in infant formula.

  1. Page 8, line 304. “The conducted studies indicate the superiority of human food over modified milk due to only trace amounts of squalene in infant formulas.” I disagreed with this statement. Firstly, I’m not aware of any scientific consensus about squalene as the essential nutrient in baby food and their recommended daily uptake. Moreover, as indicated in this study, the level of squalene in breast milk is highly variable, with 1/3 of samples contained squalene below LOQ. So it is improper to claim breast milk as better baby food because it contain highly variable amount of squalene that has unknown nutritional benefits.

Response:

The authors agree with the comment of the Reviewer. Squalene, although it has some proven health-promoting properties, there is still little research on the content of this nutrient in breast milk and its health-promoting properties on a child. The sentence has been modified. “The conducted studies have shown that the content of squalene in breast milk varies depending on the mother's age, lactation period and body weight and its relation to body height (BMI index). Although the majority of breast milk samples contained squalene, 33% of them still contained squalene to an unmeasurable value (below the lower limit of detection of squalene in fat). Although the World Health Organization recommends breastfeeding as long as possible and emphasizes the high nutritional value of mother's milk due to the unique composition (the content of proteins, polyunsaturated fatty acids, oligosaccharides, immune factors, enzymes and hormones), so far no unambiguous standards have been established. on the content of cholesterol precursors, including squalene, in breast milk and the resulting pro-health properties. The study was aimed at checking the squalene content in human milk and what potential factors may influence its formation in human milk.

Round 2

Reviewer 1 Report

The Authors have improved their manuscript significantly, still, there are some aspects to be further clarified:

L31-32: "feeding the mother food"- please, reconsider, it has no sense in this form

L103: "feeding" should be replaced by "breastfeeding" (please, chack and correct this throughout the entire manuscript)

L304-306: the intention of the reviewer was not to encourage dietary recommendations on intake of squalene-rich food, but to draw the attention of the Authors upon the fact that if mothers consume such foods, then it will have an impact on the squalene content of their milk. Please, emphasize in the manuscript that "Based on a nutritional interview with women, none of them included the characteristic food products that are sources of squalene (amaranth, olive oil, fish oils)."

Author Response

Dear Reviewer,

We would like to thank you for the positive review of our work. Please be advised that we have made the following additions:

1.    "Feeding the mother food" has been changed to "breastfeeding"
2.     "Feeding" has been changed to breastfeeding in the place designated by the reviewer and in other places where it should be.
3.     Thank you for your attention. This sentence was included in the manuscript.

The authors are convinced that the manuscript in its current form is appropriate and may be published in International Journal of Environmental Research and Public Health.

Reviewer 2 Report

This is the revised manuscript by Pietrzak-Fiećko about the investigation of squalene concentration in human breast milk. All of my concerns have been addressed by the authors. So I recommend to accept this manuscript at its present form.

Author Response

Dear Reviewer,

We would like to thank you very much for the positive review of our work: The occurrence of squalene in human milk and infant formula. 

The authors are convinced that the manuscript in its current form is appropriate and may be published in International Journal of Environmental Research and Public Health.